# ExpeSQL: An Experience-Guided Decompositional Search Framework for Text-to-SQL

## Abstract

Large language models have advanced Text-to-SQL, yet enterprise deployment remains challenging due to complex, evolving schemas, domain shift, privacy constraints, and latency/cost budgets. We introduce **ExpeSQL**, a zero-shot, open-source–compatible framework that enables self-evolving SQL generation through experience-guided refinement. ExpeSQL decomposes questions using schema-aware reasoning, generates verifiable sub-SQLs, and aggregates candidates via result-based filtering and majority voting. When errors occur, a self-critique module performs diagnostic backtracking, storing structured remedies in a persistent memory. These experiences are replayed in future rounds to prevent repeated mistakes—enabling continuous improvement without parameter updates. On BIRD-dev, ExpeSQL achieves 67.5% execution accuracy with open-source models, reducing token generation by up to 87% and inference latency by 96% compared to Alpha-SQL at similar accuracy. This superior accuracy-efficiency trade-off establishes ExpeSQL as a new paradigm for deployable, self-improving Text-to-SQL systems in dynamic real-world environments.

## 1 Introduction

Large language models (LLMs) have revolutionized Text-to-SQL systems (Liu et al., 2025a; Zhang et al., 2025), enabling non-expert users to query databases through natural language and significantly lowering the barrier to data access. While promising, the practical deployment of LLM-based Text-to-SQL systems in enterprise environments remains challenging (Cohere et al., 2025). Real-world databases are often large, schema-complex, and subject to frequent updates and domain shifts—conditions under which standard LLMs exhibit poor robustness, generating syntactically valid but semantically incorrect queries due to hallucinated joins, misaligned columns, or ambiguous user intents. Moreover, enterprise applications demand strict data privacy, low-latency responses, and minimal operational overhead, all of which are difficult to achieve with current approaches. To improve accuracy, existing methods typically rely on either training (TR) (e.g., supervised fine-tuning or reinforcement learning) (Liu et al., 2025b) or test-time scaling (TTS) (Li et al., 2025a; Yang et al., 2025b).

**TR methods**, such as Reasoning-SQL (Pourreza et al., 2025), leverage domain-specific data and advanced training objectives (e.g., GRPO) to achieve high performance—up to 72.29 execution accuracy on the BIRD dev set (Li et al., 2023) with a 14B open-source model. However, TR-based systems face critical barriers to enterprise adoption: they require extensive labeled training data, incur high retraining costs when schemas evolve, and often fail to generalize to unseen domains or new databases. More critically, fine-tuning typically involves sending sensitive SQL and schema information to third-party services or internal training pipelines, raising serious data leakage risks in regulated environments.

**TTS Methods.** To address these limitations, TTS has recently emerged as a compelling alternative to fine-tuning (Guan et al., 2025). For instance, CHASE-SQL (Pourreza et al., 2024) leverages three strategies—divide-and-conquer, query planning, and online synthetic example generation—to produce multiple candidate SQL queries, which are then selected by a fine-tuned agent. However, like many TTS approaches, it depends on closed-source APIs, incurring high operational costs and raising privacy concerns in enterprise settings.

Alpha-SQL (Li et al., 2025a) stands out as a notable exception: a purely test-time scaling method built entirely on the open-source Qwen2.5-Coder-7B model (Hui et al., 2024), achieving an execution accuracy of 66.8% on the BIRD development set—surpassing many fine-tuned baselines. In Alpha-SQL, the Text-to-SQL task is formulated as a Monte Carlo Tree Search (MCTS) (Browne et al., 2012) process, with seven predefined actions. Starting from the root node, each action generates a new node, forming a path to a leaf node that corresponds to one candidate SQL query—completing a single rollout. After 24 rollouts, the final SQL is selected via majority voting based on the consistency of execution results.

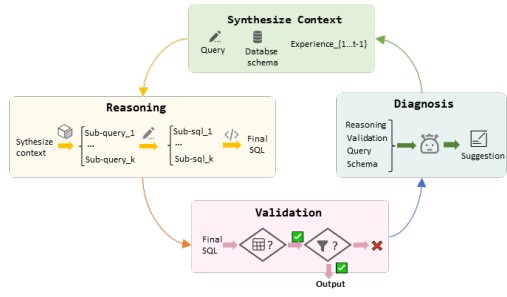

Figure 1: Reasoning loop of ExpeSQL.

Despite its high accuracy, Alpha-SQL suffers from significant inefficiencies—namely high latency and excessive token generation. Through careful implementation and analysis of their codebase, we identify three key factors contributing to this inefficiency:

1. Leaf nodes sharing the same parent inherit identical reasoning traces (i.e., chains of thought), leading to redundant SQL generation. Even across different parents, partial reasoning paths are frequently reused, reducing diversity and amplifying redundancy.

2. The node and SQL quality are represented solely by scalar scores. Although these scores are backpropagated along the path after each rollout to guide future node selection, the search process fails to leverage richer historical information—such as which segments of prior reasoning were likely erroneous—to improve search efficiency.

3. The method requires all 24 rollouts to complete before termination, lacking a flexible early-exit mechanism based on convergence or confidence.

Closely related, MCTS-SQL (Yuan et al., 2025) also adopts MCTS for SQL generation and incorporates a self-evaluation-and-refinement loop to enhance efficiency. However, its design is oversimplified, achieving only 53.0% execution accuracy on the same model (Qwen2.5-Coder-7B), significantly underperforming Alpha-SQL. This gap highlights the need for a more effective and efficient TTS framework.

Building upon these insights, we argue that ideal, production-ready Text-to-SQL systems (Cheng et al., 2025) should satisfy three criteria:

1. **Zero-shot:** Rely solely on algorithmic enhancements to strengthen the model's intrinsic reasoning capabilities, ensuring generalization across domains and question types.

2. **Open-source compatible:** Operate on open-source models in an "off-the-shelf" manner, avoiding API calls and mitigating data privacy risks.

3. **Efficient inference:** Minimize both latency and token consumption while preserving high accuracy.

To this end, we propose **ExpeSQL**, a novel framework that integrates divide-and-conquer, Best-of-N sampling, and self-reflection (Renze & Guven, 2024). As illustrated in Figure 2, the workflow of our method comprises the following stages.

- **Schema Linking:** This step is necessary because the values in a database may be numerous, it is generally impossible to input all of them into the model due to the limitation of the prompt length (Shkapenyuk et al., 2025).

- **Divide-and-conquer:** The core idea of divide-and-conquer is decomposing the questions into several simple sub-questions and solve them one-by-one (Smith, 1985). This approach has been proven to be effective in several fields, including Text-to-SQL (Pourreza et al., 2024; Wang et al., 2023).

- **SQL Selection:** Selecting the SQL query in a pool of candidates by majority voting.
- **Self-critique:** Evaluating the selected SQL query and decide if another iteration is necessary.
- **Error Diagnosis and Remediation:** When the self-critique step identifies an error, this agent analyzes the reasoning trajectory to pinpoint root causes, generates targeted remediation actions, and stores the diagnostic results in a long-term memory for future reasoning reconstruction.

ExpeSQL introduces a closed-loop reasoning architecture that combines intra-node diversity preservation, inter-node consensus voting, and self-critique validation with a persistent experience repository. This enables the system to not only generate SQL candidates but also learn from its own reasoning traces, progressively reducing error recurrence over refinement rounds. The key contribution of our approach includes:

1. We propose ExpeSQL, a novel experience-driven, self-refining framework for Text-to-SQL that integrates decompositional reasoning, execution feedback, and multi-round refinement in a unified pipeline.

2. We design a self-critique agent and a diagnostic agent that leverage the divide-and-conquer structure to enable fine-grained, modular validation and precise root-cause analysis, significantly enhancing both accuracy and interpretability in Text-to-SQL systems.

3. We introduce a dual filtering mechanism (intra-node diversity preservation and inter-node consensus voting) and a structured experience repository $E$ that enables the system to learn from past reasoning traces and prevent error recurrence.

4. Experimental results demonstrate that our method achieves a strong balance between effectiveness and efficiency, delivering high accuracy with controlled computational cost.

With open source models, our approach obtained an execution accuracy of 67.5% on BIRD dev set and achieved a good trade-off between effectiveness and efficiency.

## 2 METHOD: EXPESQL

### 2.1 OVERVIEW

We formalize a decompositional reasoning space for Text-to-SQL: given the database schema $D$ and a natural-language query $q$, the Schema Scoping and Feature Identification Agent first identifies the relevant tables and columns, producing a textual schema rationale $p$ that is incorporated into subsequent reasoning steps.

$$p = \text{Agent}_{\text{scope}}(q, \mathcal{D}) \in \mathcal{P} \tag{1}$$

Next, the Decomposition Agent breaks $q$ into a set of independent sub-questions $S = \{s_i\}$. For each sub-question $s_i$, the Subquery Translation Agent generates a corresponding sub-SQL query $\sigma_i$, which is executed by the Execution Agent to yield a result $r_i$. These intermediate results are used to construct a final SQL query $\Sigma$ that integrates all sub-SQL queries, completing a full divide-and-conquer reasoning cycle.

$$
\begin{aligned}
\Sigma &= \text{Compose}\Big( \big\{ \text{Execute}\big(\text{Translate}(s_i)\big) \big\}_{i=1}^{|S|} \Big), \\
S &= \text{Decompose}(q)
\end{aligned}
\tag{2}
$$

Our framework operates sequentially over $T$ refinement rounds, with each round guided by a long-term experience repository $E$ that stores historical reasoning traces and remediation knowledge. In each round $t$, $M$ independent reasoning nodes execute the complete pipeline in parallel. Crucially, within each node, the final SQL generation process produces $K$ candidate queries to enhance solution space coverage.

Each node then applies **intra-node result-based filtering**: the $K$ candidates are grouped by their execution results, and the two most frequent distinct outcomes are retained. From each of these two

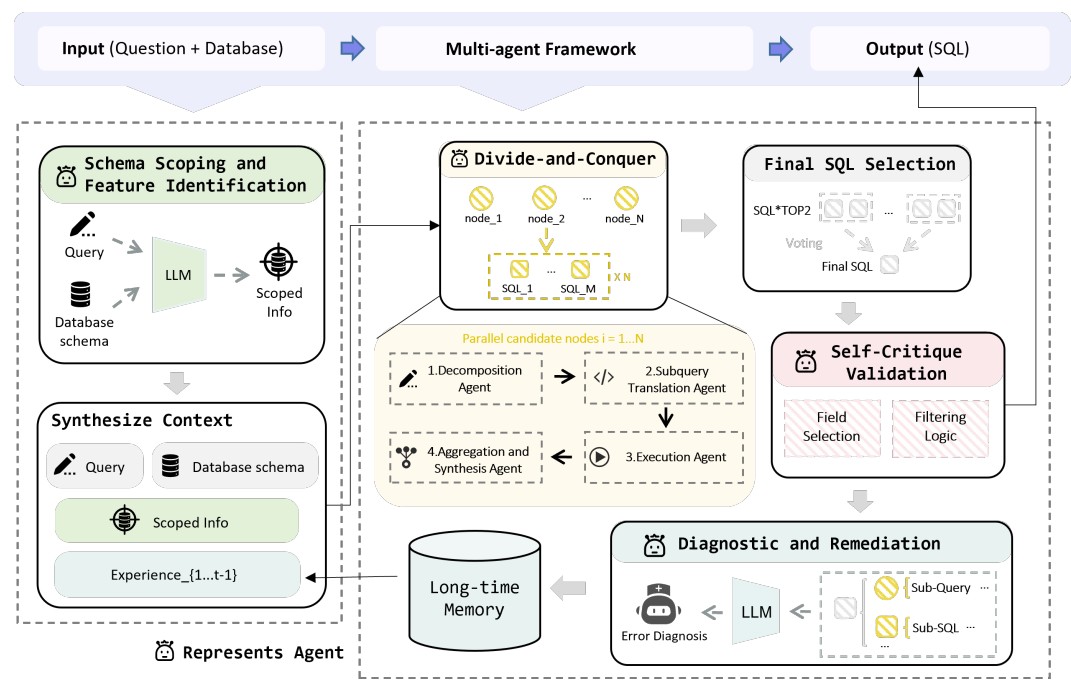

Figure 2: Overview of ExpeSQL: A closed-loop, experience-driven Text-to-SQL system. Each round involves $M$ parallel nodes generating and filtering SQL candidates via decompositional reasoning. Inter-node majority voting yields $\Sigma_t$, which is validated and critiqued. If valid, $\Sigma_t$ is returned as the final answer. Otherwise, root causes $\mathcal{E}_t$ and remedies $\mathcal{R}_t$ are extracted and stored in experience repository $E$ which guides future rounds, enabling self-refinement over $T$ iterations.

result groups, the shortest SQL is selected—yielding exactly two high-quality, diverse candidates per node.

This results in a total of $2M$ candidate SQLs across all nodes. The system then performs **inter-node majority voting** over these $2M$ candidates: the SQL whose execution result appears most frequently is selected as the round-$t$ output $\Sigma_t$; in case of a tie, the shortest such SQL is chosen.

$$\mathcal{C}_t : \text{ all candidate SQLs in round } t,$$
$$\Sigma_t \in \arg\max_{\sigma \in \mathcal{C}_t} \#\big\{ \sigma' \in \mathcal{C}_t \ : \ \text{Exec}(\sigma') = \text{Exec}(\sigma) \big\}, \tag{3}$$
$$\text{tie-break: pick the shortest such } \sigma.$$

The **Self-Critique Validation Agent** evaluates $\Sigma_t$ along two axes: (i) *field selection correctness* and (ii) *filtering correctness*. If both pass, $\Sigma_t$ is emitted as the final output. Otherwise, the **Diagnostic and Remediation Agent** receives the full reasoning chain $\langle q, p, S, \{\sigma_i\}, \{r_i\}, \Sigma_t \rangle$, performs backtracking to localize root causes $\mathcal{E}_t$ (e.g., decomposition bias in $s_k$ or translation/logic error in $\sigma_k$), and issues repair suggestions $\mathcal{R}_t$.

$$\text{Accept}(\Sigma_t) \iff \mathcal{V}_{\text{field}}(\Sigma_t) \wedge \mathcal{V}_{\text{filter}}(\Sigma_t) \tag{4}$$

$$\big(\mathcal{E}_t, \ \mathcal{R}_t\big) = \text{Diagnose}(\langle q, \ p, \ S, \ \{\sigma_i\}, \ \{r_i\}, \ \Sigma_t \rangle) \tag{5}$$

These structured artifacts-including $p$, $s_i$, $\sigma_i$, $r_i$, $\Sigma_t$, self-critique judgments, error taxonomy, and remediation actions-are persisted into $E$. In subsequent rounds, $E$ is systematically traversed to inform decomposition strategies, validate candidate solutions, and apply previously recorded remediation actions.

$$\text{Let } \Phi_t = \langle q, \mathcal{D}, p, S, \{\sigma_i\}, \{r_i\}, \Sigma_t \rangle,$$
$$\text{Update experience repository:} \tag{6}$$
$$E \leftarrow E \cup \{\Phi_t, \mathcal{E}_t, \mathcal{R}_t\}.$$

The process continues sequentially until a valid SQL is produced or the maximum round $T$ is reached. The final output $\Sigma^*$ is the last valid SQL generated, ensuring progressive refinement grounded in historical experience.

## 2.2 AGENTS

ExpeSQL employs an agentic Text-to-SQL pipeline (Figure 3) (Li et al., 2025b), where specialized LLM agents handle schema scoping, decomposition, translation, execution, and validation. This design ensures modularity, execution-level verifiability, and iterative refinement through a persistent experience repository $E$ that captures errors, diagnostics, and fixes for reuse in subsequent rounds.

**Schema Scoping and Feature Identification Agent.** This agent initiates the reasoning pipeline by performing a coarse-grained relevance analysis over the database schema. Given a natural language question, it prompts the large language model to identify and output the subset of tables and their relevant columns that are potentially involved in answering the query. By focusing only on semantically related schema elements, this step effectively narrows the search space for subsequent agents, reducing noise from irrelevant fields and improving both efficiency and accuracy downstream (Chen et al., 2025).

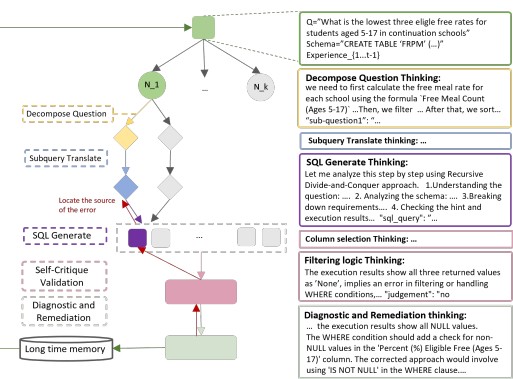

Figure 3: An example of the ExpeSQL iterative planning, diagnosis, and repair cycle.

**Agents of Divide-and-Conquer.** The **Decomposition Agent**, **Subquery Translation Agent**, and **Execution Agent** collectively realize the core reasoning loop of ExpeSQL. Given the scoped schema, the Decomposition Agent breaks the natural language query into a set of independent, semantically self-contained sub-questions $S = \{s_i\}$. For each $s_i$, the Subquery Translation Agent generates a corresponding sub-SQL $\sigma_i$ using schema-aware prompting and domain-specific templates, ensuring syntactic correctness and semantic alignment. The Execution Agent then executes each $\sigma_i$ against the database to obtain a concrete result $r_i$, forming a verifiable triplet $\langle s_i, \sigma_i, r_i \rangle$. This tight loop-decompose, translate, execute-not only enables early validation of partial logic but also generates actionable feedback (e.g., schema mismatches) that informs downstream aggregation and self-critique. By modularizing these steps, the pipeline ensures traceability, reduces error propagation, and supports parallel exploration across multiple reasoning paths (Yang et al., 2025a).

**Aggregation and Synthesis Agent.** This agent is responsible for constructing the final SQL query by integrating the intermediate sub-SQLs $\{\sigma_i\}$ and their executed results $\{r_i\}$ into a coherent, executable statement. Taking as input the set of verified sub-SQL queries and their observed outcomes, it performs semantic synthesis—aligning projections, aggregations, joins, and filters—to assemble a globally consistent SQL query $\Sigma$ that reflects the original intent (Cheng et al., 2025).

Crucially, within each reasoning node, the agent generates $K$ candidate queries through controlled perturbation of key components (e.g., join sequences, filtering conditions, aggregation scopes, and projection lists). This Best-of-$K$ generation enables exploration of alternative query structures while grounding synthesis in execution-verified sub-results. The resulting candidates are executed, and the two with the most frequent distinct outcomes are retained, selecting the shortest SQL per outcome for downstream voting.

By centralizing the synthesis process, the agent ensures syntactic completeness and semantic coherence, while enabling explicit feedback between local sub-question resolutions and global query construction—effectively bridging decomposed reasoning to final answer generation (Qu et al., 2025).

**Self-Critique Validation Agent.** This agent performs structured, execution-aware validation of candidate SQL queries before finalization (Xu et al., 2025). Given a synthesized SQL query $\Sigma$, it evaluates correctness along two orthogonal axes: (i) *field selection*, verifying the alignment of projected measures and dimensions with the natural language intent, including correct use of aggregation functions (e.g., COUNT, SUM), proper aliasing, DISTINCT semantics, and validity of selected columns with respect to the database schema; and (ii) *filtering logic*, assessing the validity of WHERE and HAVING predicates, proper handling of NULL values, and avoidance of erroneous join conditions misused as filters. The validation is grounded in both syntactic structure and semantic execution: the agent checks whether the query executes without error and whether intermediate or final results exhibit expected patterns. If $\Sigma$ passes both criteria, it is emitted as the final answer. Otherwise, the query and its full reasoning chain are forwarded to the Diagnostic and Remediation Agent for deep analysis, enabling failure-aware refinement rather than blind rejection (Zhai et al., 2025).

**Diagnostic and Remediation Agent.** While some iterative refinement methods attempt to correct logical errors (Qin et al., 2024; Shi et al., 2025; Yuan et al., 2025), they typically rely on in-place updates guided by coarse execution feedback, leading to local adjustments without systematic diagnosis of root causes. This approach risks incomplete or unstable fixes—such as over-correction (e.g., altering correct conditions) or cascading errors—due to the lack of a reconstructed reasoning trajectory. For example, in work Cao et al. (2024), which leverages powerful models such as GPT-4o and DeepSeek-R1, iterative refinement yields less than a 0.8% improvement in overall accuracy.

By integrating with the divide-and-conquer framework, the Diagnostic and Remediation Agent adopts a self-reflective mechanism that generates fine-grained, sub-problem-level feedback—pinpointing errors in specific sub-SQL queries or decomposition steps—and stores these insights in a long-term memory module to guide a complete reconstruction of the reasoning trajectory in subsequent iterations, rather than performing in-place SQL edits. Upon receiving a failed reasoning trace, this agent performs causal backtracking to localize and characterize errors within the pipeline (Somov & Tutubalina, 2025). It takes as input the complete structured context $\langle q, S, \{\sigma_i\}, \{r_i\}, \Sigma \rangle$, reconstructs the execution flow, and identifies root causes such as decomposition bias (e.g., missing or over-partitioned sub-questions $s_k$), translation inaccuracies (e.g., incorrect joins or filters in $\sigma_k$), or logical inconsistencies (e.g., type mismatches, semantic drift). Based on this diagnosis, it generates targeted repair suggestions—such as reformulating a sub-question, adjusting join logic, or revising aggregation scope—and persists the entire artifact tuple $(s_i, \sigma_i, r_i, \Sigma, \mathcal{E}_t, \mathcal{R}_t)$ into the long-term experience repository $E$. This structured memory enables continual learning (Van de Ven & Tolias, 2019): in subsequent rounds, the system proactively retrieves relevant past failures and fixes to guide decomposition, prioritize robust paths, and avoid recurrent errors, thereby closing the loop between execution, critique, and improvement.

## 2.3 Context Engineering And Experience Learning

To facilitate context management (Mei et al., 2025), we adopt a short- and long-term memory mechanism (Zhang et al., 2023). At the beginning of each iteration, an **action node** is created to encapsulate the full divide-and-conquer reasoning trace. This node includes the decomposed sub-questions $S = \{s_i\}$, their corresponding sub-SQLs $\{\sigma_i\}$, and the execution results $\{r_i\}$—organized as a sequence of verifiable reasoning steps, forming the short-term memory of the current round. Additionally, the action node stores metadata from the current iteration, including the final SQL query, its execution outcome, the self-critique result, and, if validation fails, the diagnostic output from causal backtracking.

When self-critique fails and a new iteration is initiated, the current action node becomes a child of the previous action node, forming a temporally ordered chain. This hierarchical linking across iterations constitutes the system's long-term memory, preserving historical reasoning traces and error recovery contexts. At the start of each new cycle, this long-term memory is traversed, and relevant segments are selectively incorporated into the context to guide subsequent reasoning. The memory is reconstructed in a sequential format:

$$\text{round}_1 : \big(s_i, \sigma_i, r_i, \Sigma, \mathcal{E}_t, \mathcal{R}_t\big) \xrightarrow{\text{failure}} \text{round}_2 : \big(\dots\big) \xrightarrow{\text{failure}} \cdots \xrightarrow{\text{success}} \text{round}_k$$

Agents that leverage long-term memory—including the **Schema Scoping and Feature Identification Agent**, the **Decomposition Agent**, **Subquery Translation Agent**, and the **Aggregation and Synthesis Agent**—use this accumulated experience to inform schema relevance analysis, subproblem decomposition, and query synthesis, respectively.

## 2.4 PREPROCESSING

We adopt the schema-linking methodology of Talaei et al. (2024); Li et al. (2025a). For each database, we first preprocess textual column values using MinHash (Datar et al., 2004) to produce compact signatures, which we store locally. Given a user query, we extract salient keywords with a LLM and apply locality-sensitive hashing to efficiently retrieve candidate values from the precomputed signatures. We then filter candidates using edit-distance and semantic-similarity thresholds, where semantic similarity is computed with bert-large-uncased (Devlin et al., 2018). The resulting matches are incorporated into the database-schema prompt, thereby providing contextual grounding for the LLM during SQL generation.

## 3 EXPERIMENTS

**Datasets and Metrics.** In this work, we report and compare model performance on two widely used datasets: the BIRD (Li et al., 2023) and Spider (Yu et al., 2018) development sets, which contain 1,534 and 1,034 question-SQL pairs, respectively, spanning multiple databases across diverse domains. Notably, BIRD presents a higher level of complexity compared to Spider, making it a more challenging benchmark that better discriminates the effectiveness of different approaches, as demonstrated in the Table 1 and Table 4. For TTS methods, evaluation is typically conducted along two dimensions: performance gains and computational overhead. For the former, we adopted the commonly used Execution Accuracy (EX) which indicates the portion of the SQL queries which has the same execution results as the gold SQL query. We further compare our model with Alpha-SQL—the strongest open-source, fine-tuning-free method, comparable to ours—in terms of the number of tokens generated per question, highlighting our efficiency improvements.

**Implementation details.** We use instruction-tuned variants from the Qwen-Coder series, including Qwen2.5-Coder-14B-Instruct and Qwen2.5-Coder-32B-Instruct, as well as the more recent Qwen3-Coder-30B-Instruct (Hui et al., 2024; Team, 2025). For the Best-of-N framework, we generate three reasoning nodes per iteration. During majority voting, each node produces eight candidate SQL queries to enable robust aggregation. The maximum number of iterations is set to three, allowing the model up to two opportunities to refine and correct its output if errors are detected in prior steps.

## 4 RESULTS

**BIRD dataset.** As shown in Table 1, fine-tuning significantly improves model performance on the BIRD dataset. Nevertheless, ExpeSQL outperforms several existing methods—even those based on powerful closed-source models. For instance, using the Qwen3-Coder-30B model, our approach achieves higher accuracy than CHESS-SQL (which uses GPT-4-Turbo) and Distillery (which uses GPT-4o). When training is applied to open-source models, only XiYan-SQL with M-Schema representation and Reasoning-SQL—trained via reinforcement learning—surpass ExpeSQL in performance. Notably, when XiYan-SQL adopts DDL-based schema encoding (the same representation used in our method), its performance drops below that of ExpeSQL when both are instantiated with the Qwen2.5-Coder-32B model.

In comparison with other methods in the same category—i.e., zero-shot approaches using open-source models—our method demonstrates clear advantages, substantially outperforming MCTS-SQL, ROUTE, and Distillery. The relatively small number of methods in this category reflects, to some extent, the inherent difficulty of achieving strong performance without any training.

When compared to Alpha-SQL, our method lags behind across all three model variants, though the performance gap narrows as the base model improves. While there remains a gap in execution accuracy, TTS methods are typically evaluated along two dimensions: computational cost and model

Table 1: Comparison of state-of-the-art Text-to-SQL methods on the BIRD dev set.

| Method | Inference Model | Selection Model | Zero-shot | Open-Source | EX (%) |
|---|---|---|---|---|---|
| CHESS-SQL (Talaei et al., 2024) | Deepseek-Coder-33B | GPT-4-Turbo | No | No | 65.0 |
| Distillery (Maamari et al., 2024) | GPT-4o | — | No | No | 67.2 |
| CHASE-SQL (Pourreza et al., 2024) | Gemini-1.5-Pro | Gemini-1.5-Flash | No | No | 73.0 |
| XiYan-SQL (Liu et al., 2025b) | Unreported | Unreported | No | No | 73.3 |
| DAIL-SQL (Gao et al., 2023) | GPT-4 | Majority Voting | Yes | No | 55.9 |
| SuperSQL (Li et al., 2024a) | GPT-4 | Majority Voting | Yes | No | 58.5 |
| MAC-SQL (Wang et al., 2023) | GPT-4 | Iterative Refinement | Yes | No | 59.4 |
| Gen-SQL (Shi et al., 2025) | GPT-4 | Iterative Refinement | Yes | No | 59.8 |
| RSL-SQL (Cao et al., 2024) | GPT-4o | Iterative Refinement | Yes | No | 67.2 |
| DTS-SQL (Pourreza & Rafiei, 2024) | DeepSeek-7B | — | No | Yes | 55.8 |
| SFT CodeS (Li et al., 2024b) | CodeS-15B | — | No | Yes | 58.5 |
| ROUTE (Multi-task + FT) (Qin et al., 2024) | Qwen2.5-Coder-14B | Iterative Refinement | No | Yes | 60.9 |
| CHESS-SQL (Talaei et al., 2024) | Deepseek-Coder-33B | LLaMA3-70B | No | Yes | 61.5 |
| XiYan-SQL@DDL (Liu et al., 2025b) | Qwen2.5-Coder-32B | Qwen2.5-Coder-32B | No | Yes | 62.3 |
| XiYan-SQL@M-Schema (Liu et al., 2025b) | Qwen2.5-Coder-32B | Qwen2.5-Coder-32B | No | Yes | 67.0 |
| Reasoning-SQL (Pourreza et al., 2025) | Qwen2.5-Coder-14B | — | No | Yes | 72.3 |
| MCTS-SQL (Yuan et al., 2025) | Qwen2.5-Coder-7B | Iterative Refinement | Yes | Yes | 53.6 |
| ROUTE (Multi-task only) (Qin et al., 2024) | Qwen2.5-Coder-14B | Iterative Refinement | Yes | Yes | 56.3 |
| Distillery (Maamari et al., 2024) | Llama-3.1-405B | — | Yes | Yes | 59.2 |
| Alpha-SQL* (Li et al., 2025a) | Qwen2.5-Coder-14B | Majority Voting | Yes | Yes | 64.1 |
| Alpha-SQL* (Li et al., 2025a) | Qwen2.5-Coder-32B | Majority Voting | Yes | Yes | 64.4 |
| Alpha-SQL* (Li et al., 2025a) | Qwen3-Coder-30B-A3B-Instruct | Majority Voting | Yes | Yes | 68.2 |
| **ExpeSQL (Ours)** | Qwen2.5-Coder-14B | Iterative Refinement | **Yes** | **Yes** | **61.4** |
| **ExpeSQL (Ours)** | Qwen2.5-Coder-32B | Iterative Refinement | **Yes** | **Yes** | **63.5** |
| **ExpeSQL (Ours)** | Qwen3-Coder-30B-A3B-Instruct | Iterative Refinement | **Yes** | **Yes** | **67.5** |

*EX: Execution accuracy on BIRD dev set. * Here we report results obtained using our own evaluation script, which we believe provides a more reasonable assessment; see Appendix A.2 for detailed justification.*

performance. As validated in subsequent experiments, our approach achieves significant reductions in computational overhead and inference latency, offering a highly favorable trade-off between efficiency and performance in practical deployment scenarios.

Another observation worth noting is that all methods employing *Iterative Refinement*, except for RSL-SQL based on GPT-4o, achieve relatively limited performance. For instance, ROUTE—even with fine-tuning on the Qwen2.5-Coder-14B model—still underperforms compared to ExpeSQL. This suggests that simple error detection followed by direct SQL refinement is an overly simplistic strategy with significant limitations.

In contrast, our method does not directly modify the generated SQL. Instead, it identifies the root cause of the error and, in the subsequent iteration, leverages prior experience to reformulate the query understanding and generation process. This more principled approach to iterative reasoning leads to substantially improved performance.

**Spider dataset.** Overall, due to the relatively simpler nature of the tasks, the performance gap among different methods on the Spider dataset is smaller compared to that on BIRD. ExpeSQL with the Qwen2.5-Coder-14B model outperforms DAIL-SQL and ZeroNL2SQL, both of which are based on GPT-4. Notably, despite sharing similar components—such as multi-task learning and iterative refinement—our method achieves significantly better performance on Qwen2.5-Coder-14B than the non-fine-tuned ROUTE, demonstrating the effectiveness of our design choices.

**CHESS-SDS dataset.** The CHESS-SDS dataset is a carefully curated subset of the BIRD development set, introduced in the work Talaei et al. (2024) to emphasize complex semantic parsing and schema reasoning challenges. As shown in Table 2, our method not only achieves high execution accuracy on this dataset, but also significantly reduces computational cost and inference latency compared to Alpha-SQL. Specifically, our approach reduces token generation by 87% and cuts inference latency by 96%. This dramatic improvement stems from the high degree of parallelism in our algorithm—except for the self-critique and error diagnosis stages, all other steps are executed in parallel. In contrast, Alpha-SQL only parallelizes the candidate expansion phase. As a result, our

Table 2: Comparison of Baseline LLMs on the CHESS-SDS Set

| Model | EX (%) | Tokens/Query | Time/Query (s) |
|-------|--------|--------------|----------------|
| Deepseek-R1 | 50.3 | – | – |
| GPT-4o | 53.7 | – | – |
| Gemini-2.0-Flash-Thinking-Exp | 60.8 | – | – |
| Qwen2.5-Coder-32B | 49.0 | 282 | 8 |
| + Alpha-SQL | 58.5 | 111778 | 1628 |
| **+ ExpeSQL** | 60.6 | 14510 | 66 |
| Llama-3.1-8B | 21.1 | – | – |
| **+ ExpeSQL** | 45.0 | – | – |

method achieves highly efficient inference while maintaining strong performance, outperforming all models in the comparison except for Gemini-2.0-Flash-Thinking-Exp, where it is slightly behind.

We further evaluated our method on Llama-3.1-8B (Team, 2024), achieving a substantial improvement of approximately 24% over the baseline. This demonstrates the strong generalizability and plug-and-play capability of our approach. Detailed experimental settings are provided in Section A.3.

## 5 CONCLUSION AND LIMITATIONS

We presented ExpeSQL, a zero-shot, experience-driven Text-to-SQL framework that enables self-evolving reasoning through structured decomposition, execution feedback, and experience-guided refinement. By storing diagnostic knowledge from failures, ExpeSQL improves over time without retraining, achieving strong accuracy on BIRD-dev while reducing token usage by 87% and latency by 96% compared to Alpha-SQL.

Nonetheless, several limitations point to future work: (1) SQL diversity is constrained by fixed decomposition paths—exploring alternative reasoning strategies could enhance solution space coverage; (2) intermediate candidates discarded during filtering may contain valuable partial insights, suggesting opportunities for richer experience logging; (3) correctness validation assumes tight alignment between natural language and SQL, and errors in this alignment may lead to false rejections. Improving robustness in this step remains critical.

Despite these challenges, ExpeSQL establishes a new paradigm for deployable, adaptive Text-to-SQL systems—demonstrating that intelligent, iterative refinement grounded in structured experience can bridge the gap between research and real-world database interaction.

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

## A APPENDIX

### A.1 ABLATION STUDIES

In this section, we conduct an ablation study to quantitatively evaluate the contribution of each component in our proposed framework. Table 3 summarizes the performance improvements on the BIRD-dev dataset using different methodologies.

The baseline model, Qwen32B-Coder-32B equipped with preprocessed schema information, achieves an accuracy of 53.3%. The inclusion of schema linking further improves performance by 1.9 percentage points, resulting in an accuracy of 55.2%. This gain highlights the importance of effectively identifying and aligning relevant database elements with the natural language question, reducing noise from irrelevant schema components.

Table 3: Progressive performance on BIRD-dev

| Method | Accuracy (%) |
|---|---|
| Qwen32B-Coder-32B + Preprocessed Schema | 53.3 |
| +Schema Linking | 55.2 (+1.9) |
| +Divide-and-Conquer | 59.1 (+3.9) |
| +Self-Reflection | 62.2 (+3.1) |
| +Best-of-N | 63.5 (+1.3) |

Incorporating the Divide-and-Conquer strategy leads to a substantial improvement of 3.9 percentage points, increasing accuracy to 59.1%. This significant gain underscores the effectiveness of decomposing complex questions into manageable sub-problems, enabling more accurate and modular reasoning.

Further enhancement is observed with the addition of self-reflection, which boosts the accuracy to 62.2%, a gain of 3.1 percentage points over the previous step. This indicates that allowing the model to iteratively refine its predictions through self-assessment can lead to substantial gains in accuracy.

Finally, the integration of the Best-of-N technique yields a marginal but notable improvement, increasing the accuracy to 63.5% (+1.3 percentage points). The Best-of-N approach, which selects the best output from multiple attempts, demonstrates its utility in enhancing overall performance, albeit at a diminishing rate compared to earlier enhancements.

Overall, these results highlight the incremental benefits of each additional component, demonstrating the importance of carefully designed strategies in achieving optimal performance on challenging datasets such as BIRD-dev.

### A.2 ON THE TREATMENT OF EMPTY QUERY RESULTS IN EVALUATION

In our attempt to reproduce the results of AlphaSQL, we observed a discrepancy between our initial evaluation and those reported by the authors. Upon obtaining their evaluation script, we successfully reproduced their results and found that their protocol automatically discards SQL predictions yielding empty results during majority voting, falling back to the next most frequent non-empty result. While this adjustment enables faithful reproduction of their published numbers, we argue that such a design implicitly assumes—often incorrectly—that all valid SQL queries must return non-empty results. In real-world applications, however, it is entirely plausible for a semantically correct query to return an empty result (e.g., "find customers with 10+ orders last month" in a newly launched business). We also examined several representative Text-to-SQL methods, including those based on iterative refinement or majority voting (Xie et al., 2025) (Pourreza & Rafiei, 2023) (Wang et al., 2023) (Li et al., 2024a), and found no evidence of such empty-result filtering in their public implementations or evaluation protocols. We therefore adopt a more principled evaluation protocol: we only filter out SQL queries that fail to execute or raise parsing/execution errors, but preserve those that return empty results upon successful execution. To ensure a fair and realistic comparison, we report the performance of AlphaSQL on the BIRD development set using our evaluation protocol.

### A.3 EFFICIENCY AND PERFORMANCE ANALYSIS

**Experiment Settings.** We measure the token generation cost and inference latency of both models on the CHESS-SDS (Talaei et al., 2024) dataset. For the experiments involving Alpha-SQL and ExpeSQL, we deploy Qwen32B-Coder-32B locally using vLLM (Kwon et al., 2023) on four Nvidia

GeForce RTX 4090 GPUs, each with 24GB of memory. During evaluation, we process one question at a time and record the total execution time and the total number of generated tokens per question. The final results are reported as averages over all questions.

It is important to note that for Alpha-SQL, the reported latency does not include the time required for majority voting. Specifically, timing stops immediately after all candidate SQL queries are generated. Therefore, the actual end-to-end inference time would be longer than reported. In contrast, for ExpeSQL, majority voting is an integral part of the inference pipeline, and its computational cost is included in the total latency.

Table 4: Comparison of state-of-the-art Text-to-SQL methods on the Spider dev set.

| Method | Inference Model | Selection Model | Zero-shot | Open-Source | EX (%) |
|---|---|---|---|---|---|
| DAIL-SQL (Gao et al., 2023) | GPT-4 | Majority Voting | Yes | No | 83.6 |
| ZeroNL2SQL (Fan et al., 2024) | GPT-4 | - | Yes | No | 84.0 |
| MAC-SQL (Wang et al., 2023) | GPT-4 | Majority Voting | Yes | No | 86.8 |
| SuperSQL (Li et al., 2024a) | GPT-4 | Majority Voting | Yes | No | 87.0 |
| SFT CodeS (Li et al., 2024b) | CodeS-15B | - | No | Yes | 84.9 |
| ROUTE (Multi-task + FT) (Qin et al., 2024) | Qwen2.5-Coder-14B | Iterative Refinement | No | Yes | 87.3 |
| ROUTE (Multi-task only) (Qin et al., 2024) | Qwen2.5-Coder-14B | Iterative Refinement | Yes | Yes | 80.0 |
| Alpha-SQL* (Li et al., 2025a) | Qwen2.5-Coder-7B | Majority Voting | Yes | Yes | 84.0 |
| Alpha-SQL* (Li et al., 2025a) | Qwen2.5-Coder-14B | Majority Voting | Yes | Yes | 87.0 |
| **ExpeSQL (Ours)** | Qwen2.5-Coder-14B | Iterative Refinement | **Yes** | **Yes** | **84.3** |
| **ExpeSQL (Ours)** | Qwen3-Coder-30B-A3B-Instruct | Iterative Refinement | **Yes** | **Yes** | **86.2** |

*EX: Execution accuracy on Spider dev set. \* Results for Alpha-SQL are directly taken from the original paper without re-evaluation. Due to differences in evaluation protocols, the actual accuracy under our setup may be lower than reported.*

## A.4 RELATED WORK

### A.4.1 TYPICAL PARADIGMS IN TEXT-TO-SQL

Modern Text-to-SQL systems follow a three-stage pipeline: schema linking (Shkapenyuk et al., 2025), candidate generation (Zhu et al., 2025), and candidate selection (Sheng & Xu, 2025). Schema linking aligns natural language with database elements to reduce ambiguity, evolving from rule-based matching (Lyu et al., 2025) to fine-tuned retrievers for complex schemas (Qin et al., 2024). In the second stage, SQL candidates are generated via single-path decoding (Wang et al., 2023) or multi-path reasoning (Yang et al., 2025a), often using modular decomposition (Qin et al., 2024) or diverse prompting (Lee et al., 2024). The final stage selects the best candidate using execution feedback (Li et al., 2025a) or learned rankers (Pourreza et al., 2024).

State-of-the-art systems like XiYan-SQL (Liu et al., 2025b) combine multi-path generation with fine-tuned components for high accuracy, but rely heavily on labeled data and are brittle under schema changes—limiting adaptability in dynamic environments. Moreover, many depend on closed-source LLMs (Xie et al., 2025), introducing latency, cost, and privacy concerns in production.

### A.4.2 TTS FOR TEXT-TO-SQL

To overcome the limitations of fine-tuning, recent work has explored TTS—a paradigm that enhances reasoning performance at inference time without parameter updates. TTS methods typically generate diverse SQL candidates through strategic search (Lyu et al., 2025) (e.g., self-consistency, chain-of-thought variation, or tree search) and select the final output based on execution feedback or majority voting.

MAC-SQL (Wang et al., 2023) adopts a divide-and-conquer framework to decompose complex questions into sub-questions, enabling structured reasoning over large databases. However, it relies on closed-source models such as GPT-4 and achieves only 59.4% execution accuracy on the BIRD dev set. In contrast, Alpha-SQL (Li et al., 2025a) operates entirely with open-source LLMs under the test-time scaling paradigm and requires no fine-tuning, achieving state-of-the-art performance among such methods.

Nonetheless, both approaches illustrate a persistent challenge in Text-to-SQL: the difficulty of balancing *accuracy* and *efficiency*. MAC-SQL (Wang et al., 2023), despite using powerful proprietary models, fails to deliver high accuracy, suggesting that decomposition alone is insufficient for complex reasoning. Alpha-SQL (Li et al., 2025a) achieves strong accuracy but at the cost of high computational overhead due to inefficient search strategies. This trade-off reveals a critical gap—existing methods either sacrifice performance for practicality or efficiency for accuracy—leaving a need for systems that achieve both high correctness and deployable efficiency.

To bridge this gap, we propose ExpeSQL, a highly efficient and accurate Text-to-SQL framework that achieves strong performance without sacrificing inference speed or token efficiency. By integrating decomposition-based reasoning, experience-guided search, and parallel Best-of-N sampling, our method navigates the accuracy-efficiency trade-off more effectively than prior work—enabling robust, deployable Text-to-SQL in real-world settings.

### A.5    Use of Large Language Models for Writing Assistance

We confirm that large language models were used solely for language polishing and editorial refinement during the preparation of this manuscript. All technical contributions, conceptual insights, and algorithmic designs are the exclusive work of the authors.

---

**Algorithm 1** Experience-Guided Divide-and-Conquer Text-to-SQL with Iterative Refinement

---

**Require:** Database schema $\mathcal{D}$, Natural language query $q$, Max refinement rounds $T$, Number of parallel nodes $M$, Candidates per node $K$
**Ensure:** Final SQL query $\Sigma^*$
1: Initialize experience repository $E \leftarrow \emptyset$                    ▷ Stores past traces & remedies
2: **for** $t = 1$ to $T$ **do**
3:     SCHEMASCOPING$(q, \mathcal{D}) \rightarrow p$                    ▷ Relevant tables/columns
4:     DECOMPOSITION$(q, p, E) \rightarrow S = \{s_i\}$                    ▷ Sub-questions
5:     **for** each node $m = 1$ to $M$ **in parallel do**
6:         **for** each sub-question $s_i \in S$ **do**
7:             SUBQUERYTRANSLATION$(s_i, p, \mathcal{D}) \rightarrow \sigma_i$
8:             EXECUTION$(\sigma_i, \mathcal{D}) \rightarrow r_i$                    ▷ Intermediate result
9:         **end for**
10:         AGGREGATIONANDSYNTHESIS$(S, \{\sigma_i\}, \{r_i\}, p, E) \rightarrow \{\Sigma_t^{(k)}\}_{k=1}^K$        ▷ Generate $K$ SQL queries
11:         **Intra-node filtering:**
12:         Group $\{\Sigma_t^{(k)}\}$ by execution result $r^{(k)}$
13:         Retain top-2 most frequent **distinct** results
14:         **for** each of the two result groups **do**
15:             Select shortest SQL $\Sigma_{t,m}^{(1)}, \Sigma_{t,m}^{(2)}$
16:         **end for**
17:     **end for**
18:     Collect all node outputs: $\mathcal{C}_t = \{\Sigma_{t,m}^{(1)}, \Sigma_{t,m}^{(2)}\}_{m=1}^M$
19:     **Inter-node majority voting:**
20:     Group $\mathcal{C}_t$ by execution result
21:     Select most frequent result; break ties by shortest SQL
22:     $\Sigma_t \leftarrow$ corresponding SQL query
23:     **Self-Critique Validation**$(\Sigma_t, q, \mathcal{D})$                    ▷ Uses $\mathcal{D}$ and $q$
24:     **if** VALIDATEFIELDSELECTION$(\Sigma_t, q, \mathcal{D}) \wedge$ VALIDATEFILTERINGLOGIC$(\Sigma_t, q, \mathcal{D})$ **then**
25:         **return** $\Sigma^* \leftarrow \Sigma_t$                    ▷ Final output
26:     **else**
27:         $\Phi_t \leftarrow \langle q, \mathcal{D}, p, S, \{\sigma_i\}, \{r_i\}, \Sigma_t \rangle$
28:         DIAGNOSTICANDREMEDIATION$(\Phi_t) \rightarrow$ root causes $\mathcal{E}_t$ and remedies $\mathcal{R}_t$
29:         Update $E \leftarrow E \cup \{\Phi_t, \mathcal{E}_t, \mathcal{R}_t\}$
30:     **end if**
31: **end for**
32: **return** $\Sigma^* \leftarrow$ last valid $\Sigma_t$ (or $\emptyset$ if none)

---

