# OpenReview forum: "ExpeSQL: An Experience-Guided Decompositional Search Framework for Text-to-SQL"
_ICLR.cc/2026/Conference — Submitted to ICLR 2026_

### Official Review · Reviewer_sJAv · 2025-10-26

**Soundness:** 2
**Presentation:** 3
**Contribution:** 1
**Rating:** 2
**Confidence:** 4

**Summary:**

This paper targets the deployment of Text-to-SQL systems in enterprise environments where databases are often large, schema-complex and subject to frequent updates and domain shifts.
The authors propose ExpeSQL, which combines divide-and-conquer decomposition, Best-of-N sampling, and iterative refinement with an experience repository that stores diagnostic feedback from failed queries.
On BIRD-dev, ExpeSQL achieves 67.5% accuracy (vs Alpha-SQL's 68.2%) while reducing token generation by 87% and inference latency by 96%.

**Strengths:**

1. The paper aims to address deployment challenges in enterprise Text-to-SQL systems. Its focus on zero-shot, open-source compatibility is well-motivated and practically relevant.

2. The experience repository design is sensible: by storing structured traces at the sub-question level  rather than just final SQL outcomes, the system can potentially perform more granular error diagnosis and targeted remediation. This decomposed memory structure aligns naturally with the divide-and-conquer framework and could enable interpretable debugging.

**Weaknesses:**

1. The paper does not clearly explain how sub-query–level critiques affect final SQL generation. Algorithm 1 shows sub-queries are executed and validated per node, but self-critique occurs only after full SQL composition. Section 2.3 mentions that agents “leverage long-term memory” to guide decomposition and synthesis, yet no example shows how stored remedies are retrieved or applied in later iterations. The impact of sub-query feedback on future reasoning remains unclear.

2. The proposed “sub-query critique” is essentially an additional diagnostic signal recorded after execution, not an independent reasoning or learning mechanism. The framework mainly combines known components, divide-and-conquer (MAC-SQL), self-reflection (Renze & Guven 2024), iterative refinement (ROUTE, MCTS-SQL, Gen-SQL), and Best-of-N sampling (SuperSQL, Alpha-SQL). Storing these diagnostic traces in an experience repository extends prior fine-grained error-analysis methods (Gen-SQL, SHARE) but does not introduce a substantively new paradigm.

3. The experiments do not isolate the effect of sub-query critique or experience storage. A key missing baseline is ExpeSQL with critique applied only to the final SQL, without sub-query analysis or experience replay, to test whether the fine-grained diagnostic signal provides measurable gains over standard self-reflection. Table 3 reports “Self-Reflection (+3.1%)” but conflates multiple factors, and the paper presents no per-iteration accuracy or retrieval-success statistics to substantiate the claimed benefits.

**Questions:**

see weakness

**Details Of Ethics Concerns:**

no concerns

---

### Official Review · Reviewer_DXGN · 2025-10-30

**Soundness:** 4
**Presentation:** 4
**Contribution:** 3
**Rating:** 6
**Confidence:** 5

**Summary:**

EXPESQL introduces a zero-shot, open-source–compatible framework that enables self-evolving SQL generation through experience-guided refinement. It decomposes questions, generates varifiable sub-SQLs and aggregates candidates via result based filtering and majority voting. In case of an error, a self Critique module performs diagnostic backtracking and stores structured remedies in a persistent
memory. This enables continuous improvement without parameter updates.

**Strengths:**

This paper achieves 67.5% execution accuracy with open-source models, reducing the token generations by up to 87% and inference latency by 96% compared to Alpha-SQL at similar accuracy. This is a highly valuable result for practical, real-world deployment. It has leveraged open-source models.

**Weaknesses:**

The Self-Critique Validation Agent verifies the alignment of projected measures and dimensions with the natural language intent which may be difficult and lead to falsely reject correct queries, triggering expensive and unnecessary refinement rounds reducing the efficiency.

**Questions:**

How robust is the system to agent hallucinations? In multi round refinement, how many rounds are usually needed.

---

### Official Review · Reviewer_xviF · 2025-10-31

**Soundness:** 3
**Presentation:** 2
**Contribution:** 2
**Rating:** 4
**Confidence:** 3

**Summary:**

This paper tackles the limitations of existing training-free Text-to-SQL approaches, which rely on long reasoning traces to ensure accuracy but suffer from high latency, while efficiency-oriented methods typically sacrifice correctness. To achieve a better trade-off between accuracy and efficiency, the authors propose ExpeSQL, an experience-guided multi-agent framework that decomposes complex natural language queries using a divide-and-conquer strategy and progressively refines them through Best-of-N selection, self-critique, and a reasoning experience cache. Experimental results show that ExpeSQL significantly reduces token generation and inference latency while maintaining accuracy comparable to Alpha-SQL, establishing a strong balance between performance and efficiency.

**Strengths:**

* **Well-Motivated Objective.** While most existing works either focus solely on accuracy or sacrifice accuracy to improve efficiency, this paper explicitly aims to balance both speed and quality, addressing a key practical limitation of current Text-to-SQL systems.
* **Novel Framework Design.** ExpeSQL introduces a multi-agent architecture that combines divide-and-conquer reasoning, self-evaluation, and iterative refinement. The reasoning traces are cached across iterations to reduce redundant generation, enabling more efficient self-evolution.

**Weaknesses:**

* **Limited Efficiency Comparison.** Although Section 1 discusses Alpha-SQL’s inefficiency and MCTS-SQL’s limited accuracy, the experiments only report token cost comparisons with Alpha-SQL on a single open-source model. Even when testing on Llama 3.1-8B, the paper omits token and latency data. The overall efficiency relative to other fast test-time frameworks remains unclear, and the generalization of efficiency gains across model scales is insufficiently demonstrated.
* **Missing Ablation on efficiency gain from Long-Term Memory.** The proposed reasoning-path caching with long-term memory is a central contribution, yet there is no quantitative ablation analyzing its benefit. Additional experiments—such as the distribution of refinement loops per task and the token savings with vs. without caching—would strengthen the justification of this design.
* **Uneven Benchmark Comparison.** The paper’s motivation emphasizes balancing accuracy and efficiency between MCTS-SQL and Alpha-SQL. However, Table 1 reports MCTS-SQL results only on Qwen2.5-Coder-7B, whereas Alpha-SQL and ExpeSQL use larger 14B and 32B models. A fair comparison on the same backbone is necessary to substantiate the claimed improvements.
* **Presentation Could Be Improved.** Since this work focuses on the accuracy–efficiency trade-off, visualizing the results in a 2D scatter plot (e.g., x-axis = token cost, y-axis = accuracy) would make the trade-off clearer. Moreover, presenting results by model family rather than mixing different backbones in one table would improve readability and fairness.

**Questions:**

* The authors note that Spider is relatively simple and therefore skip its evaluation, but Spider 2.0 [1]—released recently—includes more realistic and complex queries. How does ExpeSQL perform on Spider 2.0 compared to Alpha-SQL, MTCS-SQL and other recent baselines?

[1] Lei, Fangyu, et al. "Spider 2.0: Evaluating language models on real-world enterprise text-to-sql workflows." *arXiv preprint arXiv:2411.07763* (2024).

---

### Official Review · Reviewer_FTD2 · 2025-11-03

**Soundness:** 2
**Presentation:** 2
**Contribution:** 1
**Rating:** 4
**Confidence:** 3

**Summary:**

This paper proposes ExpeSQL, a zero-shot Text-to-SQL system that decomposes input questions into sub-questions, generates sub-SQLs, and refines them using a self-critique and diagnostic module. The method operates without fine-tuning and leverages an experience repository to improve over time. On the BIRD benchmark, ExpeSQL achieves 67.5% execution accuracy, outperforming previous zero-shot baselines while being significantly more efficient in token usage and latency.

**Strengths:**

- Engineering strength: Well-structured modular framework combining decomposition, subquery voting, self-correction, and memory replay.
- Competitive performance: Matches or exceeds prior zero-shot systems (e.g., Alpha-SQL) on BIRD and Spider, while reducing compute cost.
- Comprehensive baseline comparison: Evaluated against strong SOTA methods (XiYan-SQL, Reasoning-SQL, CHESS, CHASE, etc.) across BIRD, Spider, and CHESS-SDS.

**Weaknesses:**

- Limited novelty beyond prior agent-style LLM systems
The architecture draws heavily on design patterns already explored in other domains (e.g., self-refinement, agent modularity, error memory in program synthesis and math QA). While novel in the Text-to-SQL setting, the contribution is largely an application of existing ideas rather than a conceptual advancement.
- Insufficient ablation and diagnostic analysis
The ablation study is limited to a few components and only reported on one dataset. Core modules such as self-critique, experience replay, and inter-node filtering are not evaluated in isolation. The system's iterative improvement claims lack supporting data or analysis.
- No interpretability or reasoning trace evidence
The paper does not include SQL examples, error case studies, or reasoning path comparisons that could help readers understand how and why the method improves correctness. This undermines the claim that ExpeSQL improves reasoning quality.
- Missing broader benchmark coverage
There is no evaluation on Spider 2.0 (the most realistic benchmark for multi-query workflows) or on multi-turn dialogue datasets like CoSQL or SParC. This limits insight into generality and real-world applicability.

**Questions:**

The core ideas are not new in the broader LLM literature, and the paper misses an opportunity to provide insights into how its components work or improve reasoning. A more thorough ablation, analysis of reasoning behavior, and demonstration of generality across diverse scenarios would have strengthened the contribution.

---

### Meta-Review · Area_Chair_pRM7 · 2026-01-03

**Summary:**

Reviewers raised concerns about the framework's limited novelty as an application of existing LLM agent patterns without conceptual advancements, insufficient ablations isolating components like self-critique and experience replay, lack of interpretability through SQL examples or error analyses, uneven benchmark comparisons with missing evaluations on Spider 2.0 or multi-turn datasets, and potential efficiency reductions from false query rejections, informing a suggested rejection due to inadequate rigor and innovation.

**Reviewer Concerns:**

No rebuttal is present in the document, so none of the concerns have been addressed; all remain outstanding, including novelty, ablations, benchmark coverage, interpretability, and efficiency justifications.

**Reviewer Scores:**

Without a rebuttal or further discussion, reviewers would likely retain their original scores.

---

### Decision · Program_Chairs · 2026-01-26

Reject